# Coupling relationship between cold chain logistics and economic development: A investigation from China

**Ruhe Xie[1], Hong Huang[1]***, **Yuan Zhang[1,2], Peiyun Yu[3]**

**1** School of Management, Guangzhou University, Guangzhou, China, **2** School of Management, Guizhou University of Commerce, Guiyang, China, **3** School of Management, Guangzhou College of Commerce, Guangzhou, China

* Huanghong0929@e.gzhu.edu.cn

## Abstract

This paper builds an evaluation index system, uses the entropy weight method (EWM) to decide the weights and, based on the coupling coordination degree model (CCDM), it systematically studies the coupling relationship between Chinese cold chain logistics and the Chinese economy from 2010 to 2019. It performs a grey relational analysis (GRE) to explore the main factors influencing the coordinated development of the two. The results show that the coupling coordination degree between the two presents a steady upward trend, and their coupling relationship has been upgraded from 'coordination' to 'good coordination'. They also indicate that the added value in the tertiary industry, the per capita gross domestic product (GDP), and household consumption levels are the main factors affecting the development of cold chain logistics, while the per capita cold storage capacity, the turnover of road cold chain freight, and the volume of human-power employed in cold chain logistics are the main factors affecting economic development. This study makes suggestions to support the coordinated development of cold chain logistics and economy, and provides a scientific basis for further research.

## 1. Introduction

As the rapid development of the global economy intensifies the process of urbanization, and consumer requirements and expectations regarding the quantity and quality of fresh food continue to rise, cold chain logistics becomes the fundamental prerequisite to fulfilling consumer requirements [1]. The term refers to a systematic process that ensures that refrigerated and frozen food is always stored and transported in an appropriately low-temperature environment at all stages, from production to consumption, to guarantee quality and reduce waste and loss to a minimum [2–4]. According to the report of the China Cold-chain Logistics Association [5], the market demand for cold chain logistics in China increased by 17.5% in 2019, and the market size is expected to reach RMB 915 billion (US $140 billion) in 2025, with an average annual growth rate of 18% that should provide tremendous potential for the overall increase of the market economy.

It is worth noting that the relationship between cold chain logistics and economic development is becoming increasingly close [6]. The development of the former drives the latter by

**Funding:** This research was funded by the National Social Science Foundation of China (grant number 17BJY102).

**Competing interests:** We have declared that no competing interests exist.

allowing macro-economic control, innovating diversified economic development modes, stabilizing prices and farmer income, and guaranteeing food safety [1, 7–9]. In turn, creating and supporting a solid industrial infrastructure, providing ample financial support, stimulating market demand potential, and enhancing industry service professionalization can spur the development of cold chain logistics [10–12]. However, if the overall economic development cannot generate sufficient demand for cold chain logistics capacity, further investment in human, financial, and other resources will only result in a waste of governmental and social resources. Reversely, if the input of government and social resources fails to meet the developmental needs of cold chain logistics, the consequences on the safety of the transported fresh agricultural products and the quality of life of consumers will undermine the overall economic development and the vitality of the sector in particular, as neither will be able to reach its full potential [10]. In other words, cold chain logistics and the economy are very likely to benefit from mutual, combined, coordinated growth, which is the optimal option to maximize the profits of stakeholders. However, current studies mainly focus on the impact of cold chain logistics on the economy, tend to neglect the reverse influence, and fail to further enquire on their coordinated development. Therefore, our research makes important contributions to our understanding of their interaction.

This study analyzes the interaction between Chinese cold chain logistics and the economy. Nowadays, with the help of information technology (e.g., blockchain, Internet of Things, big data, cloud computing, AI), cold chain logistics is in a better position to assist suppliers in managing orders, controlling inventory, meeting consumer demand for convenience and timely delivery, and benefit from economies of scale [6, 13, 14]. In terms of benefits generation, economic growth encourages the formation of innovative processes for cold chain logistics to optimize the supply chain, which increases consumer demand for products reliant on cold chain logistics and stimulate further development of the industry.

Therefore, this study revolves around the correlation between cold chain logistics and economic development and examines their coupling relationship. It is important to capture the coordination mechanism among various factors of cold chain logistics and economic development theoretically and empirically and to explore effective methods to measure the two.

This paper aims to build an evaluation framework with an index system that can evaluate the coordinated development level of cold chain logistics and economy, in order to analyze and measure the coupling relationship between cold chain logistics and economic development, to determine the main factors affecting their coordinated development, and to suggest specific measures to enhance their coordination. The paper is structured as follows. Section 2 outlines trends in existing literature on the interaction between cold chain logistics and the economy. Section 3 constructs the index evaluation system and presents the research methods, namely EWM, CCMD, and GRA. Section 4 analyzes the coordinated development of cold chain logistics and the economy. Section 5 proceeds with an analysis and discussion of the influencing factors affecting their coordinated development, and outlines policy recommendations, Section 6 concludes the paper and Section 7 briefly expounds on research limitations and suggests directions for future research.

## 2. Literature review

In an era marked with enormous changes in consumption habits and patterns, scholarly studies focus on the sustainable development of logistics and the economy, but without elaborating specifically on the coordinated development relationship between cold chain logistics and the economy. The metropolitan economy contributes to the increase in demand for logistics and the investment in infrastructure, thus enhancing the coordinated development of the metropolitan economy and logistics [15, 16]. More significantly, there is a positive interaction between the metropolitan economy and logistics [10]. The global economy and improved living standards have

diversified the demand for logistics. The development of cold chain meets consumer demand by ensuring timely delivery of a wide range of products, from fresh food delivery to vaccines shipment. Cold chain is an extremely important sector of the logistics industry and has made a substantial contribution to a developing Chinese economy in recent years [1].

Regarding the cold chain logistics operation mode, Liu et al. [17] created a Joint Distribution-Green Vehicle Routing Problem model and concluded that joint distribution in cold chain logistics can balance economic and environmental benefits. Rodrigues et al. [18] analyzed the factors affecting the efficiency of third-party logistics suppliers of refrigeration services in Brazil and made suggestions to tackle challenges in the supply chain to improve company performance. Some scholars considered the relevant technologies to optimize the operation of cold chain logistics. Defraeye et al. [19] suggested ways to optimize the pre-cooling technology for fresh agricultural products before they are placed into refrigerated containers and decrease the heat load in the cold chain logistics chain to reduce energy consumption. Dao et al. [20] studied the insulation materials used for vaccine transport containers and found that the basic algorithm developed to simulate complex heat transfer processes can also be a fast and economical tool for screening suitable phase change materials to reduce transportation costs. Li et al. [21] applied the Internet of things technology to real-time monitoring of safety and quality parameters of perishable products, such as temperature and humidity during transportation, in the cold chain logistics transportation process. Other researchers probed into the influencing factors of cold chain logistics. Zhang et al. [22] discussed the impact of government subsidies towards reducing carbon emissions and relevant trading policies on regional cold chain logistics, and provided suggestions for the government and enterprises to support their joint effort to form a low-carbon economy and an energy-conservation mentality. In times of global economic growth, Korean logistics enterprises chose cold chain logistics with higher added value to enter overseas markets, and pointed out that the development of an emerging logistics market cannot do without the cooperation between central and local governments and private enterprises, as well active strategies to encourage investment [23, 24]. In addition, social, environmental, and technological factors also influence the sustainable development of cold chain logistics [6, 21].

However, those studies seldom explore the influencing factors that indicate the development level of cold chain logistics, which are important in policymaking and innovating operation modes. In addition, the role of the economy in promoting cold chain logistics development is not fully considered.

Meanwhile, the influence of economic development on cold chain logistics has attracted significant scholarly attention. Qin and Tian [25] used the grey clustering method to analyze the development level of cold chain logistics in regions with significant agricultural production in China. The results revealed that regional differences in economic development and resource distribution affect the regional development level of Chinese agricultural cold chain logistics. The higher the pace of economic progress and the richer the resources, the higher the level of regional agricultural cold chain logistics development. Park [26] investigated the influencing factors that determine the location of global cold chain logistics hubs by employing the analytic hierarchy process (AHP) method, which revealed that the market appeal of the sector was the primary factor. Ikegawa and Tokunaga [27] studied the factors influencing the location of Japanese frozen food companies in foreign countries. They found that the geographical factors that determine a company's decision to enter foreign markets are affected by the popularity of refrigerators and by well-known cost reduction factors, such as wage levels, levels of national investment, and support from government policies. Zhang and Pang [28] estimated that the cold chain logistics joint distribution technology in developed countries is generally mature, and the level of urban economic development affects the development of urban fresh

agricultural products cold chain logistics. That is, the more prosperous the urban economy, the swifter the development of urban fresh agricultural products cold chain logistics, and their research mainly focuses on the cold chain logistics for agricultural products. Also, economic growth has boosted the consumers' disposable income and improved living standards. Consequently, consumers attached great importance to timely delivery, safety and quality of fresh agricultural products, and scholars are exploring ways to create and optimize cold chain logistics processes in terms of vehicles [6, 29], inventory strategy [30, 31] and information transmission [32, 33], In turn, all these contribute to the development of cold chain logistics.

Despite the extensive scholarly attention and thorough research on the topic, existing literature neglects several important aspects, which can enhance our understanding of the close relationship between cold chain logistics and the economy. First, in the empirical analyses of the relationship between cold chain logistics and economic development, existing research lacks a comprehensive evaluation of the development level of cold chain logistics. Current research only aims at probing the service level or benefits level of cold chain logistics and, to the best of our knowledge, not a single study attempts to appreciate the overall level of development of cold chain logistics. Second, only a few scholars have investigated the development of cold chain logistics and the economy in China, and none has approached it from a macro perspective. Third, scholars have concentrated their attention on the impact of cold chain logistics on the economy and fail to fully appreciate the correlation between economic factors and cold chain logistics, such as GDP per capita, per capita disposable income gap, and household consumption level. Instead, scholars dwell more on the indirect impact of economic development on cold chain logistics. Therefore, it is necessary to construct a reliable and applicable evaluation system to measure the coupling coordinated relationship of cold chain logistics and economic development, to analyze the main factors affecting their coordinated development, to provide useful insights, and to suggest future directions for the development of the industry and the economy.

## 3. Data sources and methodology

### 3.1. Data sources

The data used to evaluate the relationship between the coordinated development of the Chinese cold chain logistics industry and the overall economy were derived from the China Statistical Yearbook (2010–2020), the Statistical Bulletin of National Economic and Social Development, the China Cold Chain Development Report (2010–2020), the China Cold Chain Logistics Industry Market Prospect and Investment Strategy Analysis Report (2017–2020), and the China Logistics Statistics Yearbook (2010–2020).

### 3.2. Construction of the evaluation index system

The comprehensive evaluation method is generally applicable to evaluate the coordinated development of different systems [34]. Coupled with the findings of other researchers, this paper established an index evaluation system including economic development and cold chain logistics. The comprehensive economic development evaluation index system included 3 evaluation dimensions and 10 evaluation indicators, and the comprehensive cold chain logistics evaluation index system included 3 evaluation dimensions and 10 evaluation indicators, as shown in Table 1.

For the purpose of this paper, we measured the comprehensive evaluation index system of economic development with the use of the following different dimensions:

1. Economic growth level
   Economic growth level refers to the increase of economic efficiency and the optimization of

**Table 1. Evaluation index system for cold chain logistics and economic development.**

| System | Dimension | Indicator | Code |
|---|---|---|---|
| Economic development | Economic growth level | The per capita GDP (Yuan) | Y1 |
| | | Total investment in fixed assets (billion Yuan) | Y2 |
| | | The added value in the tertiary industry (billion Yuan) | Y3 |
| | | The per capita retail sales of consumer goods (Yuan) | Y4 |
| | Foreign trade level | The ratio of total import and export trade to GDP (%) | Y5 |
| | | The ratio of direct foreign investment to GDP (%) | Y6 |
| | Living standards | The per capita main food consumption (Kg) | Y7 |
| | | The per capita disposable income gap (Yuan) | Y8 |
| | | The number of Employees (ten thousand) | Y9 |
| | | Household consumption levels (Yuan) | Y10 |
| Cold chain logistics | Cold chain logistics development benefits | The ratio of cold chain road transportation revenue to total sector revenue (%) | X1 |
| | | The ratio of combined revenues of the top 100 cold chain enterprises to total sector revenue (%) | X2 |
| | Cold chain logistics development and operational capacity | Total value of food transported (billion Yuan) | X3 |
| | | The growth rate of cold chain freight transported via road networks (%) | X4 |
| | | The turnover of road cold chain freight (million tons/km) | X5 |
| | | Food cold chain logistics demand (ten thousand tons) | X6 |
| | Cold chain logistics development foundation | The volume of human-power employed in cold chain logistics (ten thousand) | X7 |
| | | The growth rate of expenditure of urban cold chain logistics (%) | X8 |
| | | The per capita cold storage capacity (m$^3$/person) | X9 |
| | | The growth rate of the overall number of refrigerated vehicles (%) | X10 |

economic structure. GDP is one of the most commonly suitable indicators to estimate the level of economic development [10, 34–36], yet more indicators can be found in other evaluation systems due to different evaluation purposes. For example, the investment in social fixed assets affects direct foreign investment and the sustainability of economic growth, while the stable progress of the tertiary industry and the increase in per capita retail sales of consumer goods allow the optimization of the economy's structure [37–39]. Thus, we selected four indicators to evaluate the level of economic growth: the per capita GDP, total investment in fixed assets, the value added in the tertiary industry, and the per capita retail sales of consumer goods.

2. Foreign trade level
Foreign trade level plays an important part in driving economic development and can function as an index that captures the degree of economic activity in China. The inflow and outflow of foreign capital can reflect prospects of and potential for economic development [37, 40, 41]. Therefore, we selected two indicators to measure foreign trade level: the ratio of total import and export trade to GDP and the ratio of direct foreign investment to GDP.

3. Living standards
Living standards are another important indicator to measure economic development [42, 43]. The per capita main food consumption, the per capita disposable income gap, and household consumption levels can be used as indicators, the number of employees reflects the activity of the market economy, and the development of the market economy affects people's living standards. Improved values for these indicators suggest narrower social and income inequalities, which can shape a stable environment that facilitates economic growth. Hence, we used these four indicators to assess living standards in China.

To create the comprehensive evaluation index system of cold chain logistics we used the following different dimensions:

1. Cold chain logistics development benefits
   This dimension was measured by two indicators: the ratio of road cold chain transportation revenue to total revenue, and the ratio of the combined revenues of the top 100 cold chain enterprises to total sector revenue. The road transportation revenue is the core component of cold chain logistics revenue. What's more, given that cold chain logistics enterprises are an important element in the cold chain logistics industry, their profitability reflects the benefit for the industry.

2. Cold chain logistics development and operational capacity
   The transportation capacity and demand for perishable food or cold chain logistics services reflect the development and operational capacity of the sector. Because the larger part of turnover originates in the distribution of fresh agricultural products, the number of operators and total freight volume rise together with the increasing consumer demand for fresh agricultural products. The higher the turnover of goods, the faster the distribution speed, the steeper the growth of freight volume, and the stronger the transportation capacity of cold chain logistics. According to a report from the China Cold-chain Logistics Association [5], the availability of other modes of transportation notwithstanding, road transportation comprised 80% of the total transportation volume of refrigerated and frozen foods. For this reason, four indicators were used to determine the development and operation capacity of cold chain logistics: total value of food transported the growth rate of cold chain freight transported via road networks, the turnover of road cold chain freight, and food cold chain logistics demand.

3. Cold chain logistics development foundations
   The foundations for the development of cold chain logistics refer to the factors that support the development of cold chain logistics activities. Cold chain products are common in urban distribution, and distribution providers take into consideration the perishability and short storage time of cold chain products, they have high requirements for time and infrastructure in the transportation process. Hence, substantial investment in infrastructure is a prerequisite for optimizing cold chain transportation, such as reasonable increases of the required cold storage capacity and the quantity of refrigerator trucks [1, 44–47]. At the same time, the increasing demand for refrigerated and frozen foods has led to the growth of home delivery cold chain services [48]. In view of the fact that agricultural products must be stored at low temperature during transportation and any failure to comply with such requirements will compromise the quality and safety of products [49–52]. Consequently, careful planning of the entire transportation process of products from cold storage to end consumers onboard refrigerated vehicles between cold chain links is of the utmost necessity. Thus, four indicators were used to weigh the cold chain logistics development foundation: the volume of human-power employed in cold chain logistics, the growth rate of expenditure of urban cold chain logistics, the per capita cold storage capacity, and the growth rate of the overall number of refrigerated vehicles.

### 3.3. Entropy weighting method (EWM)

EWM, originally developed by Shannon [53], is used to establish objective weights for attributes or responses. The probability theory is applied to calculate uncertain information (entropy) and determine the importance of each attribute excluding decider preferences [40,

54, 55]. EWM was applied to determine the indicator weight of economic development systems and cold chain logistics systems. The application of entropy weight can effectively decrease the subjectivity of weight decisions. The specific steps of analysis were as follows:

1. Standardizing each indicator ($x_{ij}$): As the units and magnitudes of the indicators were inconsistent, and each indicator had a positive or negative impact on the system, they could not be compared directly. Standardization of the indicators' benefits eliminated the influence of units, magnitudes, and types on the subsystem [40]. Given an research problem with $m$ research objects and $n$ evaluating indicators, $a_{ij}$ is the value of the $j$th ($j$ = 1,2,3...$n$) indicator under the $i$th ($i$ = 1,2,3...$m$) object. The original data matrix $a = (a_{ij})'_{m \times n}$ was constructed. Then, the matrix $a$ was translated to $x_{ij} = (a_{ij})_{m \times n}$ with standardization. $x_{ij}$ ($0 \leq x_{ij} \leq 1$) denotes the value in the $i$th object related to $j$th evaluation indicator, as calculated with the use of Eq (1):

$$x_{ij} = \begin{cases} \dfrac{a_{ij} - \min(a_{ij})}{\max(a_{ij}) - \min(a_{ij})}; & a_{ij} : \text{positive index} \\ \dfrac{\max(a_{ij}) - a_{ij}}{\max(a_{ij}) - \min(a_{ij})}; & a_{ij} : \text{negative index} \end{cases} \quad (1)$$

2. Calculating the information entropy ($e_j$) as in Eq (2):

$$e_j = -(Inm)^{-1} \sum_{i=1}^{i=m} p_{ij} In p_{ij} \quad (2)$$

Where $p_{ij} = \dfrac{x_{ij}}{\sum_{i=1}^{i=m} x_{ij}}$ represents the ratio of the $i$th evaluation object $x_{ij}$ under the $j$th indicator. If $p_{ij}$ = 0, then $p_{ij} In p_{ij}$ = 0.

3. Calculating indicator weights ($w_j$) as in Eq (3):

$$w_j = \dfrac{1 - e_j}{\sum (1 - e_j)} \quad (3)$$

## 3.4. Coupling coordination degree model (CCMD)

The CCMD, which entails the concept of sustainable development, is significant for measuring the coordination of logistics and economic development [56]. Coupling degree is a concept introduced from and widely used in physics, but it features in other research fields too because of the similar coupling relationship between systems [57]. When subsystems within a system reach the state of coupling and coordinated development, their positive influence is significant, which is beneficial to the sustainable development of the system [58, 59]. The CCMD in physics was utilized to estimate the coupling relationship. The aim was to reflect the development level and coordination effect of cold chain logistics and the economy as a whole, and effectively measure the level of coordinated development between the two, given as follows:

1. Developing the coupling degree (C). The coupling degree indicates the extent to which the cold chain logistics system and the economic development system correlate through their respective coupling factors. Depending on the scale of the coupling degree, cold chain logistics and economic development are classified into four coupling stages, as

**Table 2. Coupling stages and judgment criteria.**

| Coupling effect | Low level coupling stage | Antagonistic stage | Fitting in stage | High level coupling stage |
|---|---|---|---|---|
| Coupling degree $C$ | [0,0.3] | [0.3,0.5] | [0.5,0.8] | [0.8,1] |

shown in Table 2. The values of C are expressed in Equation:

$$C = \frac{2\sqrt{UG}}{U+G} \tag{4}$$

Where $U = \sum_{i=1}^{i=n} w_j x_j$ is the comprehensive development index for cold chain logistics and $G = \sum_{j=1}^{j=n} w_j x_j$ is the comprehensive development index for the economy. The comprehensive development index reflects the comprehensive development level of each system.

2. Estimating the coupling coordination degree (D). The coupling coordination degree measures the extent of the interdependence, coordination, and mutual promotion between cold chain logistics and the economy; the stronger the relationship, the higher the index. The values of D are calculated with the use of Eq (5):

$$D = \sqrt{CT} \tag{5}$$

Where $T = \alpha U_i + \beta G_i$ is the comprehensive evaluation index between cold chain logistics and the economy, and $\alpha$ and $\beta$ are the weight coefficients of each system. It is assumed that cold chain logistics and the economy are equally important, hence $\alpha = \beta = 0.5$ [34, 59, 60]. Therefore: $C \in [0,1]$. $D \in [0,1]$, and the higher the value of D, the higher the coupling coordination degree. According to [34, 58, 61], types of coupling coordination degrees in the study were divided into ten levels, as shown in Table 3.

## 3.5. Grey relational analysis (GRA)

GRA, originally proposed by Deng [62], entails the quantitative description and comparison of the developmental change of the dynamics in a system and is a useful tool to construct a prediction model that explains the response dependence on input parameters [20, 63–65]. In view of the correlation and time sequence between cold chain logistics and the economy, GRA was performed to analyze the main factors influencing the coordinated development of cold chain logistics and the economy as follows:

1. Computing the correlation coefficient ($\gamma_{ij}$). $Z_i(k)$ and $Q_j(k)$ represent the standardized values of cold chain logistics and economic indicators respectively. $\rho$ is the distinguishing

**Table 3. Coupling coordination grade and criterion.**

| Coordination grade | Coordination degree | Coordination grade | Coordination degree |
|---|---|---|---|
| Extreme imbalance | [0,0.1] | Coordination | [0.5,0.6] |
| Serious imbalance | [0.1,0.2] | Basic coordination | [0.6,0.7] |
| Moderate imbalance | [0.2,0.3] | Moderate coordination | [0.7,0.8] |
| Mild imbalance | [0.3,0.4] | Good coordination | [0.8,0.9] |
| Imbalance | [0.4,0.5] | High quality coordination | [0.9,1] |

**Table 4. Coupling strength and discrimination criteria.**

| Coupling effect | Weak | Moderate | Strong | Extremely strong |
|---|---|---|---|---|
| Correlation degree $\varphi_{ij}$ | [0,0.35] | [0.35,0.65] | [0.65,0.85] | [0.85,0.1] |

coefficient and generally takes the value of 0.5 (usually $\rho \in [0,1]$) [65]. The values of $\gamma_{ij}$ are calculated by Eq (6) as follows:

$$\gamma_{ij}(Z_i(k), Q_j(k)) = \frac{\min \min |Z_i(k) - Q_j(k)| + \rho \max \max |Z_i(k) - Q_j(k)|}{|Z_i(k) - Q_j(k)| + \rho \max \max |Z_i(k) - Q_j(k)|} \quad (6)$$

2. Obtaining the grey relational matrix ($\varphi_{ij}$). The correlation coefficient $\gamma_{ij}$ is averaged according to sample size. The closeness of the relationship between the evaluation indicators of the two systems can be analyzed by comparing the values of $\varphi_{ij}$. The higher the $\varphi_{ij}$, the stronger the correlation between the two indicators and the stronger the coupling; reversely, the lower the $\varphi_{ij}$, the weaker the correlation and the coupling. The intensity of the coupling effect are scaled into four stages, as shown in Table 4. The values of $\varphi_{ij}$ are represented as Eq (7):

$$\varphi_{ij} = \frac{1}{n}\sum_{k=1}^{n} \gamma_{ij}(Z_i(k), Q_j(k)) \quad (7)$$

## 4. Results and analysis

### 4.1. EWM to calculate weight

The EWM was applied to calculate the weight of indicators reflecting the development level of cold chain logistics and economic development level. The results could be obtained via Eqs (1), (2), and (3), as shown in Table 5.

Table 5 shows that 'the ratio of total import and export trade to GDP' indicator was the most significant factor affecting the development of the economy with a percentage weight of 14.58%, followed by 'the ratio of foreign direct investment to GDP' (14.13%) and 'the added value in the tertiary industry' (10.49%) indicators. The main indicator 'food cold chain logistics demand' was the most significant factor affecting the development level of cold chain logistics with a percentage weight of 14.86%, followed by 'the growth rate of cold chain freight transported via road networks' (14.03%) and 'the ratio of combined revenues of the top 100 cold chain enterprises to total sector revenue' (13.53%).

### 4.2. Coupling relationship comprehensive assessment

The CCMD based on EWM was employed to analyze the coupling coordination of cold chain logistics and economic development. After solving Eqs (4) and (5), the results of coupling degree, coupling coordination degree, and their respective comprehensive development indicator of the two systems were obtained. Combined with Tables 2 and 3, the coupling stage and coupling coordination degree of the two was determined, as shown in Table 6 and Fig 1.

Table 6 and Fig 1 show that from 2010 to 2019 the coupling coordination degree and the comprehensive development indexes between cold chain logistics and the economy had all increased at different paces. The cold chain logistics comprehensive development index fluctuated more widely compared with the economic comprehensive development index. Moreover,

**Table 5. Index weight for cold chain logistics and economy.**

| System | Dimension | Indicator | Weight (%) |
|---|---|---|---|
| Economic development | Economic growth level | The per capita GDP (Yuan) | 9.15 |
| | | Total investment in fixed assets (billion Yuan) | 8.17 |
| | | The added value in the tertiary industry (billion Yuan) | 10.49 |
| | | The per capita retail sales of consumer goods (Yuan) | 9.42 |
| | Foreign trade level | The ratio of total import and export trade to GDP (%) | 14.58 |
| | | The ratio of direct foreign investment to GDP (%) | 14.13 |
| | Living standards | The per capita main food consumption (Kg) | 8.84 |
| | | The per capita disposable income gap (Yuan) | 8.49 |
| | | The number of Employees (ten thousand) | 7.01 |
| | | Household consumption levels (Yuan) | 9.72 |
| Cold chain logistics | Cold chain logistics development benefits | The ratio of cold chain road transportation revenue to total sector revenue (%) | 4.07 |
| | | The ratio of combined revenues of the top 100 cold chain enterprises to total sector revenue (%) | 13.53 |
| | Cold chain logistics development and operational capacity | Total value of food transported (billion Yuan) | 10.60 |
| | | The growth rate of cold chain freight transported via road networks (%) | 14.03 |
| | | The turnover of road cold chain freight (million tons/km) | 9.49 |
| | | Food cold chain logistics demand (ten thousand tons) | 14.86 |
| | Cold chain logistics development foundation | The volume of human-power employed in cold chain logistics (ten thousand) | 9.97 |
| | | The growth rate of expenditure of urban cold chain logistics (%) | 8.34 |
| | | The per capita cold storage capacity (m$^3$/person) | 7.37 |
| | | The growth rate of the overall number of refrigerated vehicles (%) | 7.76 |

the former was significantly lower than the latter before 2018, which indicates that the development rate of cold chain logistics was lower. After 2018, cold chain logistics developed rapidly and exceeded the rates of economic development. Additionally, the coupling degree (between

**Table 6. Evaluation results for coupling relationship between cold chain logistics and economy.**

| Year | Cold chain logistics comprehensive development index $U$ | Economic comprehensive development index $G$ | Coupling degree $C$ | Coupling stages | Coupling coordination degree $D$ | Coupling coordination grade |
|---|---|---|---|---|---|---|
| 2010 | 0.2033 | 0.3904 | 0.9490 | High level coupling stage | 0.5307 | Coordination |
| 2011 | 0.2881 | 0.3964 | 0.9874 | High level coupling stage | 0.5813 | Coordination |
| 2012 | 0.2405 | 0.3708 | 0.9770 | High level coupling stage | 0.5465 | Coordination |
| 2013 | 0.3642 | 0.4744 | 0.9913 | High level coupling stage | 0.6447 | Basic coordination |
| 2014 | 0.4205 | 0.4840 | 0.9975 | High level coupling stage | 0.6717 | Basic coordination |
| 2015 | 0.4011 | 0.4834 | 0.9957 | High level coupling stage | 0.6636 | Basic coordination |
| 2016 | 0.3481 | 0.5304 | 0.9782 | High level coupling stage | 0.6555 | Basic coordination |
| 2017 | 0.3969 | 0.5582 | 0.9856 | High level coupling stage | 0.6861 | Basic coordination |
| 2018 | 0.5075 | 0.5689 | 0.9984 | High level coupling stage | 0.7330 | Moderate coordination |
| 2019 | 0.7186 | 0.5954 | 0.9956 | High level coupling stage | 0.8088 | Good coordination |

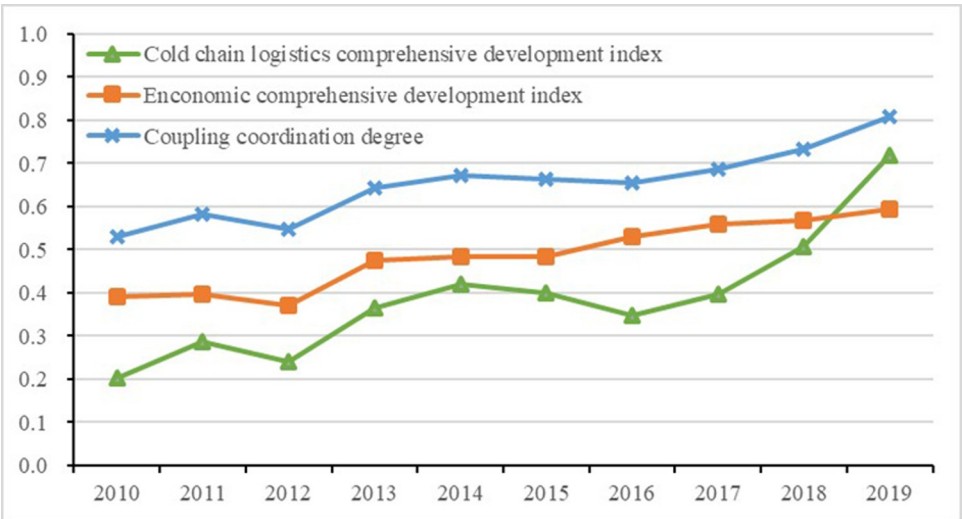

**Fig 1. Trend in coupling relationship from 2010 to 2019.**

0.9490 and 0.9956) was obviously higher than the coupling coordination degree (between 0.5307 and 0.8088), mainly because the coupling degree only measures consistency and synchronization. The lower the comprehensive development index of cold chain logistics and the economy, the higher the coupling degree [60]. Therefore, the coupling coordination degree can reflect accurately the coordinated development level of the system.

Overall, the coupling coordination degree of cold chain logistics and the economy has retained a relatively stable upward trend, as shown in Fig 1. The development of the two systems has progressed from the 'coordination' to 'good coordination' stage, as shown in Table 6. From 2010 to 2012, the coupling coordination degree fluctuated between 0.5307 and 0.5465, which shows that the development of cold chain logistics and the economy was at the 'coordination' phase. From 2013 to 2017, the coupling coordination degree fluctuated between 0.6447 and 0.6861, which suggests that the development of cold chain logistics and the economy were in the transitional phase from 'basic coordination' to 'moderate coordination'. From 2018 to 2019, the coupling coordination degree fluctuated between 0.7330 and 0.8088, which shows that the development of cold chain logistics and the economy was in the transitional stage from 'moderate' to 'good coordination'. According to the development stage changes of cold chain logistics and economic development, this study further analyzed the logic of their evolution.

From 2010 to 2012, the implementation of the *Logistics Industry Adjustment and Revitalization Plan* succeeded in making the logistics industry a sector of strategic importance and created new opportunities for the development of the cold chain logistics industry. A few new enterprises emerged, but the overall level of the industry was far from adequate and the market paid little attention to cold chain logistics.

From 2013 to 2017, as the accelerated process of urbanization increased household consumption and created an unprecedented awareness and pursuit of a healthier lifestyle, fresh foods gradually grew in popularity. National policies and industry standards were further refined. For example, the *Ministry of Finance Ministry of Commerce Notice on Central Financial Support for the Development of Cold Chain Logistics*, released in 2016, and the *Implementation Opinions on Accelerating the Development of Cold Chain Logistics to Safeguard Food Safety and Promote Consumption Upgrading*, one year after, aimed to support the development of

cold chain logistics. They stressed the urgency of increasing financial support for cold chain logistics, upgrading quality monitoring standards for perishable foods (e.g., aquatic products, meat, dairy products, and frozen foods) and completing cold chain logistics infrastructure. Then, government funds were invested in cold chain logistics enterprises, domestic and foreign cold chain companies entered the fresh food market and founded suitable distribution centers (e.g., Sinotrans, Pfizer, Swire, Metro, Walmart). Meanwhile, Chinese fresh e-commerce transactions maintained a high annual growth level of more than 30%. As the primary system of distribution of fresh products, the market has also attached great importance to cold chain logistics. The development of the market economy has further enhanced the coordinated development of cold chain logistics and the economy, although the development of cold chain logistics remained slower than the development of the economy.

From 2018 to 2019, with the improvement of cold chain infrastructure and the continuous optimization of logistics and transportation modes, there developed a growing trend to utilize multimodal transportation in cold chain fresh agricultural products distribution. Moreover, new technologies (e.g., Internet of Things, artificial intelligence, blockchain) became a vigorous driving force for the industry. Over time, cold chain logistics has entered a period of rapid development, to the effect that the comprehensive development rate of cold chain logistics is now higher than the comprehensive growth rate speed of the economy; the two have progressed first to 'moderate' and then to 'good coordination' level.

## 4.3. Analysis of main factors affecting the coupling coordination development

Based on Eqs (6) and (7), we calculated the grey relational matrix reflecting the coupling effect between cold chain logistics and the economy respectively. Combined with Table 4, the coupling strength was determined, as shown in Table 7.

**4.3.1. Analysis of economic factors affecting cold chain logistics.** Table 7 shows that the most important among the economic factors affecting cold chain logistics was 'the added value in the tertiary industry' with a maximum average correlation degree of 0.7433, followed by 'the per capita GDP' (0.7322), and 'household consumption levels' (0.7314). Their average correlations all rose to 0.65, the coupling effect was strong, the average correlation of 'ratio of total import and export trade to GDP' was the lowest one (with an average correlation of 0.584), and the coupling effect with cold chain logistics reached the 'moderate' stage.

In 2019, the added value in the logistics industry accounted for 28.08% of the total added value across the tertiary industry. As a substantial part of the logistics industry, the total

**Table 7. Matrix for the coupling effect of cold chain logistics and economy.**

|  | X1 | X2 | X3 | X4 | X5 | X6 | X7 | X8 | X9 | X10 | Avg |
|---|---|---|---|---|---|---|---|---|---|---|---|
| Y1 | 0.6743 | 0.7236 | 0.8299 | 0.4868 | 0.8961 | 0.7735 | 0.8847 | 0.5657 | 0.9400 | 0.5479 | 0.7322 |
| Y2 | 0.7968 | 0.6165 | 0.6863 | 0.5342 | 0.7268 | 0.6491 | 0.7219 | 0.5197 | 0.7881 | 0.5347 | 0.6574 |
| Y3 | 0.6554 | 0.7511 | 0.8614 | 0.4765 | 0.9326 | 0.7998 | 0.9162 | 0.5516 | 0.9496 | 0.5392 | 0.7433 |
| Y4 | 0.7136 | 0.7020 | 0.7869 | 0.4937 | 0.8387 | 0.7390 | 0.8461 | 0.5312 | 0.9234 | 0.5270 | 0.7102 |
| Y5 | 0.4733 | 0.5976 | 0.4895 | 0.8043 | 0.4743 | 0.5015 | 0.4680 | 0.7903 | 0.4799 | 0.7054 | 0.5784 |
| Y6 | 0.4637 | 0.6575 | 0.5870 | 0.6930 | 0.5585 | 0.5880 | 0.5346 | 0.7851 | 0.5343 | 0.7252 | 0.6127 |
| Y7 | 0.7174 | 0.6310 | 0.7301 | 0.5640 | 0.7484 | 0.7157 | 0.7944 | 0.5113 | 0.776 | 0.5786 | 0.6767 |
| Y8 | 0.5188 | 0.5497 | 0.5275 | 0.7721 | 0.5198 | 0.5166 | 0.5260 | 0.7780 | 0.5362 | 0.6627 | 0.5907 |
| Y9 | 0.8078 | 0.5860 | 0.6444 | 0.5351 | 0.6757 | 0.6156 | 0.6690 | 0.5099 | 0.7209 | 0.5563 | 0.6321 |
| Y10 | 0.6815 | 0.7269 | 0.8254 | 0.4812 | 0.8880 | 0.7697 | 0.8938 | 0.5538 | 0.9570 | 0.5368 | 0.7314 |
| Avg | 0.6503 | 0.6542 | 0.6968 | 0.5841 | 0.7259 | 0.6669 | 0.7255 | 0.6097 | 0.7605 | 0.5914 | 0.6665 |

revenue in cold chain road transportation accounted for 2.42% of the total revenue in the logistics industry. Hence, the growth of the tertiary industry can quickly boost the development of the logistics industry [36] and capital investment in cold chain logistics [1], while overall economic development can fuel the demand for cold chain logistics services [15, 66]. Moreover, household consumption levels reflect the strength of consumer purchasing power, and consumer spending on cold chain products affects the development of transportation and distribution businesses operating in the cold chain logistics industry.

**4.3.2. Analysis of cold chain logistics factors affecting economic development.**    Table 7 shows that 'the per capita cold storage capacity' had the strongest impact on economic development with an average correlation degree of 0.7605, followed by 'the turnover of road cold chain freight' (0.7259), and 'the volume of human-power employed in cold chain logistics' (0.7255). The average correlation degree of the indicators rose above 0.65 and the coupling effect was strong, which shows that these three factors were the main ones affecting Chinese economic development. The impact of the growth rate of cold chain freight transported via road networks was the least important one, but the correlation degree was still 0.578 and the coupling effect was at the 'moderate' stage.

Investment in transportation and logistics infrastructure restricts the development of the logistics industry and direct foreign investment, which in turn affects sustainable economic growth [10, 37, 56]. Based on the uncertain consumer demand, cold storage capacity has an adjustment effect on the uneven distribution of cold chain products in time and space in the process of cold chain logistics development. However, Chinese actual per capita cold storage capacity is less than half of that of more developed countries in Europe and America, and of Japan. Thereby, the scale of cold storage capacity limits the development level of the cold chain logistics market as well as the development of the regional economy.

Regarding the impact of road cold chain freight turnover on the economy, the relationship between the development of China's freight volume and GDP has become increasingly closer. Since the beginning of the 20[th] century, the correlation between freight turnover and GDP has increased slightly [67]. In conditions of increasingly fierce competition in the transportation market, the ratio of road cold chain transportation to sea transportation, railway transportation and air transportation rose to 88.97% in 2019, according to the China Cold-chain Logistics Association [5]. The turnover of highway cold chain freight seems to indicate that consumer demand and the demand for fresh and frozen products affect the development of the market economy. However, the impact of scattered consumers distribution, diverse demand, and inadequate transportation makes it impossible for operators to deliver small volumes and multiple batches of goods directly, resulting in an increase in the turnover of cold chain transportation and distribution costs. Furthermore, the contribution of the industry to economic development is finally weakened.

With respect to the impact of the number of cold chain logistics employees on the economy, the traditional cold chain logistics process relies more on labor force, and investment in labor force determines operational efficiency and economic benefits. With the advent of modern information and artificial intelligence, the efficiency of cold chain logistics activities was improved with the help of the Internet of things, big data, and digital empowerment precipitated immense changes in the economy [6, 68].

## 5. Policy recommendations

The relationship between Chinese cold chain logistics and economic development is a key topic of research and future government policies can draw on our findings to better tailor their intervention to meet market needs. Moreover, because China is a developing country, the

study of the coordinated development of cold chain logistics and the economy can inform decisions in other developing countries too. As the overall economy will continue to grow, the coordination and interaction between cold chain logistics and economic development appear almost inevitable. To promote the sustainable and coordinated development of Chinese cold chain logistics and the economy, we express the following recommendations:

1. Transform the cold chain logistics pattern. The increase in private disposable income and higher consumption levels have boosted total retail sales of consumer goods. Furthermore, consumption patterns of frozen and fresh foods have changed and adaptation to new market conditions is a necessity for cold chain logistics. Enterprises operating in the market should standardize fresh food distribution and set up a cold chain network to provide comprehensive, effective, timely, and regular services. In addition, the creation of a smart monitoring platform to oversee the entire process should not only allow efficient monitoring of online orders, but also make temperature and humidity levels visible and controllable. For example, across the Chinese cold chain logistics market, which was severely impacted by the COVID-19 pandemic in 2020, there emerged new models, new business types and new technologies (e.g., "fresh food e-commerce + cold chain delivery", "central kitchen + food materials cold chain distribution") for handling and transporting fresh agricultural products, which ensured the timely delivery of fresh and frozen products to the customer's door, and centralized processing of those products created more opportunities to stimulate consumption, accelerate the development of the industry, but also had a cumulating effect in driving the development of the online and digital economy.

2. Improve the cold chain logistics infrastructure. Favorable infrastructure is conducive to the successful operation of the entire network. The completion of the transportation network promotes economic development by reducing delivery time, decreasing cargo damage costs, and improving customer satisfaction. Reversely, overall economic development provides incentives to invest in transportation infrastructure. Strategies and plans to improve the infrastructure may include the following: improvement of the road transportation network; coordination and integration of the railway, sea, and air transportation networks to facilitate multimodal transport; increase cold storage capacity; upgrade refrigerated vehicles; development of a comprehensive information service system as soon as possible; optimization of the cold chain product information traceability system; integration of smart technologies to effectively plan cold chain transportation routes and select cold storage layout points.

3. Standardize the cold chain logistics guidelines. In all stages of the process (harvesting, precooling, transportation, and distribution from the place of origin to the customer's door), the control of humidity, temperature, and delivery time strongly affects product quality and safety. To meet the primary consumer demand for freshness and safety, effective management should be established to manage handover, transportation, distribution, storage, personnel, delivery tracking, product recall policy and process, documentation, and other aspects.

4. Train cold chain logistics experts. The industry relies on interdisciplinary specialist knowledge and demands a holistic approach that combines theory and practice. Hence, it is important for the sustainability and efficiency of the industry to recruit and train skillful employees who understand theory and refined technology. For example, technical and operational managers must have a working knowledge of chemistry and specific chemicals involved, such as ammonia, Freon, and Teflon, and be familiar with the operation of cold storage.

## 6. Conclusions

Based on the panel data relating to Chinese cold chain logistics and the overall economic development from 2010 to 2019, this paper examined the coupling relationship between Chinese cold chain logistics and the economy, and analyzed the main factors affecting their coordinated development. The research findings can be described as follows:

1. From 2010 to 2019, the coupling coordination degree of Chinese cold chain logistics and economic development fluctuated between 0.5307 and 0.8088, which indicates progress from 'coordination' to 'good coordination' level. Additionally, the coupling coordination degree showed a stable upward trend. In general, the synergy effect between cold chain logistics and economic development could do with improvement.

2. The coupling relationship between cold chain logistics and economic development has become increasingly close. Their coordinated development was mostly supported by the development of the economy until 2018, before cold chain logistics entered a phase of rapid development that ushered in a sustainable, coordinated development.

3. Among the evaluation indicators for economic development, 'the added value in the tertiary industry', 'the per capita GDP', and 'household consumption levels' were the main ones affecting the development of cold chain logistics (the former returning the highest average correlation degree and exerting the greatest influence).

4. Among evaluation indicators for cold chain logistics, 'the per capita cold storage capacity', 'the turnover of road cold chain freight', and 'the volume of human-power employed in cold chain logistics' were the main ones affecting economic development (the former displaying the highest average correlation degree and exerting the greatest impact).

5. There is still a long way to go before achieving satisfactory levels of coordinated, sustainable development between cold chain logistics and the economy. In light of the current situation in the industry and the main factors influencing the coordinated development of cold chain logistics and the economy, we proposed specific strategies to attain the coordinated development of the two.

The main contributions of this study are as follows:

1. The construction of an evaluation indicators system suitable to objectively and comprehensively assess cold chain logistics and economic development, which can better analyze and study the coupling relationship of the two.

2. This study adopted the EWM and CCDM to measure the development status and coupling coordination level of cold chain logistics and economic development. Both EWM and CCDM proved valid methods to measure the coordinated development of the two. Moreover, this paper fills the gap in the literature on the coupling relationship between cold chain logistics and economic development.

3. The paper articulates concrete and feasible recommendations appropriate for the development of cold chain logistics and economic development. These recommendations assist in better understanding the coupling coordination relationship between the two in terms of evaluation content, and support the optimization of the evaluation system and its application in assessing their coordinated development.

## 7. Limitations and future research directions

Chinese cold chain logistics are late-comers in an industry that benefits from longevity elsewhere (Japan, the United States, Germany, and other developed countries). With a weaker foundation and an unsatisfactory system, the industry lacks comprehensive statistical data (such as decay rates of fresh products or circulation rates of fresh agricultural products) to measure its own development, Therefore, No evaluation system of cold chain logistics can be flawless or beyond criticism. Future research can collect relevant cold chain measurement indicators more thoroughly, and expand the scope by studying the relationship between cold chain logistics and other industries, such as the financial industry or e-commerce. Lastly, national policies and economic input are often not in sync with the development of cold chain logistics. Sometimes, substantial capital investment in the industry requires time to affect the development of cold chain logistics. The temporal gap necessary to catch up with financial input for government spending to yield results invites further research.

## Supporting information

**S1 Raw data.**
(XLSX)

**S1 File.**
(DOCX)

## Acknowledgments

The authors are very grateful to anonymous reviewers for their attention to this study and useful comments.

## Author Contributions

**Data curation:** Hong Huang.

**Funding acquisition:** Ruhe Xie.

**Investigation:** Hong Huang.

**Supervision:** Ruhe Xie, Yuan Zhang, Peiyun Yu.

**Writing – original draft:** Hong Huang.

**Writing – review & editing:** Hong Huang.

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
