## [Decision Letter · Decision Letter 0]

6 Dec 2021

PONE-D-21-28480Coupling relationship between cold chain logistics and economic development: A investigation from ChinaPLOS ONE

Dear Dr. huang,

Thank you for submitting your manuscript to PLOS ONE. After careful consideration, we feel that it has merit but does not fully meet PLOS ONE’s publication criteria as it currently stands. Therefore, we invite you to submit a revised version of the manuscript that addresses the points raised during the review process.

We look forward to receiving your revised manuscript.

Kind regards,

Ming Zhang, Ph.D.

Academic Editor

PLOS ONE

Journal Requirements:

This paper is funded by the National Social Science Foundation Project “mechanism and effect evaluation of government support for the development of agricultural products cold chain logistics based on external effects（17BJY102）”

YES

Supported by supervision's project

5. Please amend your list of authors on the manuscript to ensure that each author is linked to an affiliation. Authors’ affiliations should reflect the institution where the work was done (if authors moved subsequently, you can also list the new affiliation stating “current affiliation:….” as necessary).

6. We note you have included a table to which you do not refer in the text of your manuscript. Please ensure that you refer to Table 8 & 0 in your text; if accepted, production will need this reference to link the reader to the Table.

Reviewers' comments:

Reviewer's Responses to Questions

**Comments to the Author**

1. Is the manuscript technically sound, and do the data support the conclusions?

Reviewer #1: Partly

Reviewer #2: Yes

2. Has the statistical analysis been performed appropriately and rigorously? 

Reviewer #1: Yes

Reviewer #2: Yes

3. Have the authors made all data underlying the findings in their manuscript fully available?

Reviewer #1: Yes

Reviewer #2: Yes

4. Is the manuscript presented in an intelligible fashion and written in standard English?

Reviewer #1: Yes

Reviewer #2: Yes

5. Review Comments to the Author

Reviewer #1: Highlight the innovation and value of the paper. Add scientific value or novelty in the introduction.There are some editing/grammar errors throughout the paper that need to be corrected. This paper needs a round of professional proofreading.

Reviewer #2: The topic selection of the article is novel, and the writing structure and the problems to be elaborated are clearly expressed. But I still have the following suggestions. If the author is ready to change it, I think it can be published on PLOS ONE.

1. Could you include a separate section in Part 2 called "Research gaps" and focus clearly on how research gaps develop from the different streams of literature you mention. You have briefly argued this question in the last few paragraphs of Part 2. However, it would be helpful to discuss the same issues in a more detailed way in a separate section.

2. It is important to note that the authors should give the full name when first mentioning the abbreviation.

3. Authors should proofread their paper and check for possible typos and language mistakes.

4. It’s better to provide a list of abbreviations, since there are so many used.

5. The conclusion should precede the references. Authors should carefully check the structure of paragraphs to ensure readability.

6. PLOS authors have the option to publish the peer review history of their article (what does this mean?). If published, this will include your full peer review and any attached files.

Reviewer #1: No

Reviewer #2: No

---

## [Author Response · Author response to Decision Letter 0]

28 Dec 2021

Thank you for your suggestions. I have uploaded the word documents of the answers to the experts' questions.

---

## [Decision Letter · Decision Letter 1]

14 Feb 2022

Coupling relationship between cold chain logistics and economic development: A investigation from China

PONE-D-21-28480R1

Dear Dr. huang,

We’re pleased to inform you that your manuscript has been judged scientifically suitable for publication and will be formally accepted for publication once it meets all outstanding technical requirements.

Kind regards,

Ming Zhang, Ph.D.

Academic Editor

PLOS ONE

Additional Editor Comments (optional):

Reviewers' comments:

Reviewer's Responses to Questions

**Comments to the Author**

1. If the authors have adequately addressed your comments raised in a previous round of review and you feel that this manuscript is now acceptable for publication, you may indicate that here to bypass the “Comments to the Author” section, enter your conflict of interest statement in the “Confidential to Editor” section, and submit your "Accept" recommendation.

Reviewer #1: All comments have been addressed

2. Is the manuscript technically sound, and do the data support the conclusions?

Reviewer #1: Yes

3. Has the statistical analysis been performed appropriately and rigorously? 

Reviewer #1: Yes

4. Have the authors made all data underlying the findings in their manuscript fully available?

Reviewer #1: Yes

5. Is the manuscript presented in an intelligible fashion and written in standard English?

Reviewer #1: Yes

6. Review Comments to the Author

Reviewer #1: The whole paper is very consistent with the journal. I think the revised paper is up to the standard for publication.

7. PLOS authors have the option to publish the peer review history of their article (what does this mean?). If published, this will include your full peer review and any attached files.

Reviewer #1: No

---

## [Editor Report · Acceptance letter]

17 Feb 2022

PONE-D-21-28480R1 

Coupling relationship between cold chain logistics and economic development: A investigation from China 

Dear Dr. huang:

I'm pleased to inform you that your manuscript has been deemed suitable for publication in PLOS ONE. Congratulations! Your manuscript is now with our production department. 

Kind regards, 

on behalf of

Dr. Ming Zhang 

Academic Editor

PLOS ONE